# Ecosystem Service Synergies Promote Ecological Tea Gardens: A Case Study in Fuzhou, China

Chunyi Wang [1], Mingyue Zhao [1,*], Yinlong Xu [1], Yuncheng Zhao [2] and Xiao Zhang [3]

1   Institute of Environment and Sustainable Development in Agriculture,
    Chinese Academy of Agricultural Sciences, Beijing 100081, China
2   National Meteorological Center, Beijing 100081, China
3   Key Laboratory of Land Consolidation and Rehabilitation, Land Consolidation and Rehabilitation Center,
    Ministry of Natural Resources, Beijing 100035, China
*   Correspondence: zhaomingyue@caas.cn; Tel.: +86-10-82109766

**Abstract:** Exploring the trade-off/synergy among ecosystem services (ESs) of agroecosystems could provide effective support for improving agricultural resilience for sustainable development. The construction of ecological tea gardens is emerging, aims to achieve a win-win situation for the tea industry and ecological environment protection. However, the effect of ES trade-offs/synergies on tea production is still not clear. In this study, we selected Fuzhou city, China, as a case study and explored the relationship among tea production and ESs in 2010 and 2020. Integrated Valuation of Ecosystem Services and Trade-offs (InVEST) and Intelligent Urban Ecosystem Management System (IUEMS) models were used to assess the ecosystem (dis)services, which were tea production, water yield, soil retention, net primary productivity (NPP), climate regulation, soil erosion and carbon emissions. Then, the sum of trade-off/synergy coefficients of ESs ($C_{ts}$) were defined to reveal the trade-off/synergy in tea gardens and areas except tea gardens (ETG areas). K-means clustering was used to assess the spatiotemporal change of traditional tea garden and ecological tea garden, reflecting the effect of ecological tea garden construction. The results showed that: (1) the high-value areas of tea production were mainly distributed in Lianjiang County, with yields up to 3.6 t/ha, and the low-value areas in Yongtai County, with yields from 0.1–1.0 t/ha. Other ESs showed spatial heterogeneity. (2) The trade-offs in ETG areas intensified from 2010 to 2020, with $C_{ts}$ decreasing from −0.28 to −0.73, and the synergy in tea garden was at risk of decline, with $C_{ts}$ decreasing from 4.46 to 1.02. (3) From 2010 to 2020, 96.72% of traditional tea gardens (Area I) were transformed into ecological tea gardens (Areas IV and V). (4) Further, we classified the tea garden into five zones based on tea yield, with Zone I as the low tea yield areas and Zone V as the highest. From Zone I to Zone V, the $C_{ts}$ increased from 2.6 to 7.5 in 2010, and from 1.9 to 6.5 in 2020, respectively. These results demonstrate the effectiveness of the construction of ecological tea gardens in Fuzhou and provide a reference for subsequent studies on the ESs of tea gardens and governance of ecological tea gardens.

**Keywords:** ecosystem services; trade-off; synergy; ecological tea garden; clustering

## 1. Introduction

Ecosystem services (ESs) are benefits that humans derive directly or indirectly from ecosystems [1] and refer to the natural environmental conditions and utilities that form and sustain human existence [2]. The relationship between ESs include trade-off (negative relationship), synergy (positive relationship) and compatibility (no significant relationship) [3]. Trade-off means that the increase or decrease of a certain ecosystem service leads to the decrease or increase of another [4,5]. Synergy refers to the relationship between the simultaneous decrease or increase of two or more ESs [6,7]. For example, globalization and economic development, leading to urbanization and deforestation, have promoted the provision of agricultural products, timber and other supply services, but have also

exacerbated soil erosion, carbon loss and other environmental risks. The trade-off/synergy relationships of ESs are ubiquitous and have received more attention and exploring the relationship between different types of ESs has become one of the core topics in the field of landscape ecology [8–10].

Studies have shown that the trade-off/synergy relationship between ESs could be identified spatiotemporally through the measurement of the correlation coefficient, RMES and spatial mapping [11–14]. Additionally, zoning analysis of ESs can better show the different embodiments of the relationship between ESs in comprehensive zoning [15–17]. Researchers have attempted to explore the impact of ESs on crop production [18,19]. Some studies showed that there is a prevalent trade-off relationship between crop yield and regulating/cultural ESs [20,21]. This is usually caused by pursuing crop yield in the process of agricultural management while sacrificing the quality of the agroecological environment [22,23]. Traditionally, trade-offs within agroecological systems exacerbate socioecological problems and lead to human–land conflicts. Agroecological construction, rising in recent years, has led to the formation of internal synergistic relationships that create sustainable and positive interactions, especially for agricultural products whose quality is closely related to its growing environment. Agroecological construction aims to achieve a win-win situation for the agricultural industry and ecological environment protection. However, there is still a lack of evidence to show that trade-off/synergy relationships change during the ecological construction process and how this impacts crop yield [24,25].

Ecological tea garden construction is a typical and popular mode of agroecological construction and has been promoted in many areas globally. During the cultivation of tea plantations, light, moisture, soil conditions and topography are necessary for its yield formation [26]. Due to the delicate management requirements of tea production, tea cultivation has higher requirements for the ecological environment compared to other crops [27]. In particular, the production of high-quality tea requires a safe and ecologically sound environment. Ecological tea gardens are a kind of green tea garden with sustainable utilization [28]. An ecological tea garden can maintain the stability of the ecological environment of tea gardens and obtain high-yield and high-quality tea products by restoring, maintaining and strengthening the management practice of ecological harmony [29]. With the socio-economic development and the spread of the concept of environmental protection in recent decades, Chinese demand on high-quality tea has increased. The concept of ecological tea gardens is gradually being recognized and the practice is boosted. Fuzhou city has a favorable climate for tea production and has been an important tea production base in China, and many demonstrations of an ecological tea garden have been constructed in recent years [30,31]. Under the leadership of the government and the positive response of farmers, the proportion of ecological tea gardens in Fuzhou city reached more than 70% by 2021 [32].

In addition, as an independent ecosystem, ESs of tea gardens have not been adequately studied previously [33,34]. In order to explore the trade-off/synergy relationship among tea production and ESs, we analyze whether the synergies of ESs promote tea production after ecological tea garden construction. We selected Fuzhou, China as the study area and selected seven important ecological (dis)services related to tea planting [35–40], which are tea production, water yield, soil retention, net primary productivity (NPP), climate regulation, soil erosion and carbon emissions. To be specific, this study aims to: (1) map tea production and ESs in 2010 and 2020 of Fuzhou, China; (2) compare the trade-off/synergy relationships between ESs in tea garden areas and areas except tea gardens (ETG areas); (3) discover the trend of synergistic relationship evolution within tea gardens through clustering analysis; and (4) explore whether ES synergies promote tea production. To satisfy the local development goals and the interests of different stakeholders, the results of this study can deepen the understanding of ES interactions and scientific support for sustainable ecosystem management of tea gardens.

## 2. Methods and Data Sources

### 2.1. Study Areas

Fuzhou is located in eastern Fujian Province and the lower reaches of the Minjiang River (25°15′–26°9′N, 118°08′–120°31′E), with a total land area of 11,862 km² and a sea area of 10,573 km² (Figure 1). Fuzhou geomorphology is a typical estuary basin. The basin is surrounded by mountains between 600 to 1000 m above sea level, and the whole area is high in the west and low in the east. The mountainous and hilly areas account for 72.86% of the total area of Fuzhou. Fuzhou is located in a typical subtropical monsoon climate, with an average annual precipitation of 900–2100 mm, an average annual temperature of 20–25 °C and a frost-free period of 326 d. Tea grows better in a warm and moist environment. Studies show that an altitude of about 600–800 m is most suitable for tea growth [41]. Fuzhou has unique natural resource conditions for cultivating tea, which provides the most suitable ecological environment for tea growth in terms of light, temperature, water and heat. Jasmine tea, a specialty of Fuzhou, is a variety of tea made from green tea and jasmine flower scenting, which is loved by many people.

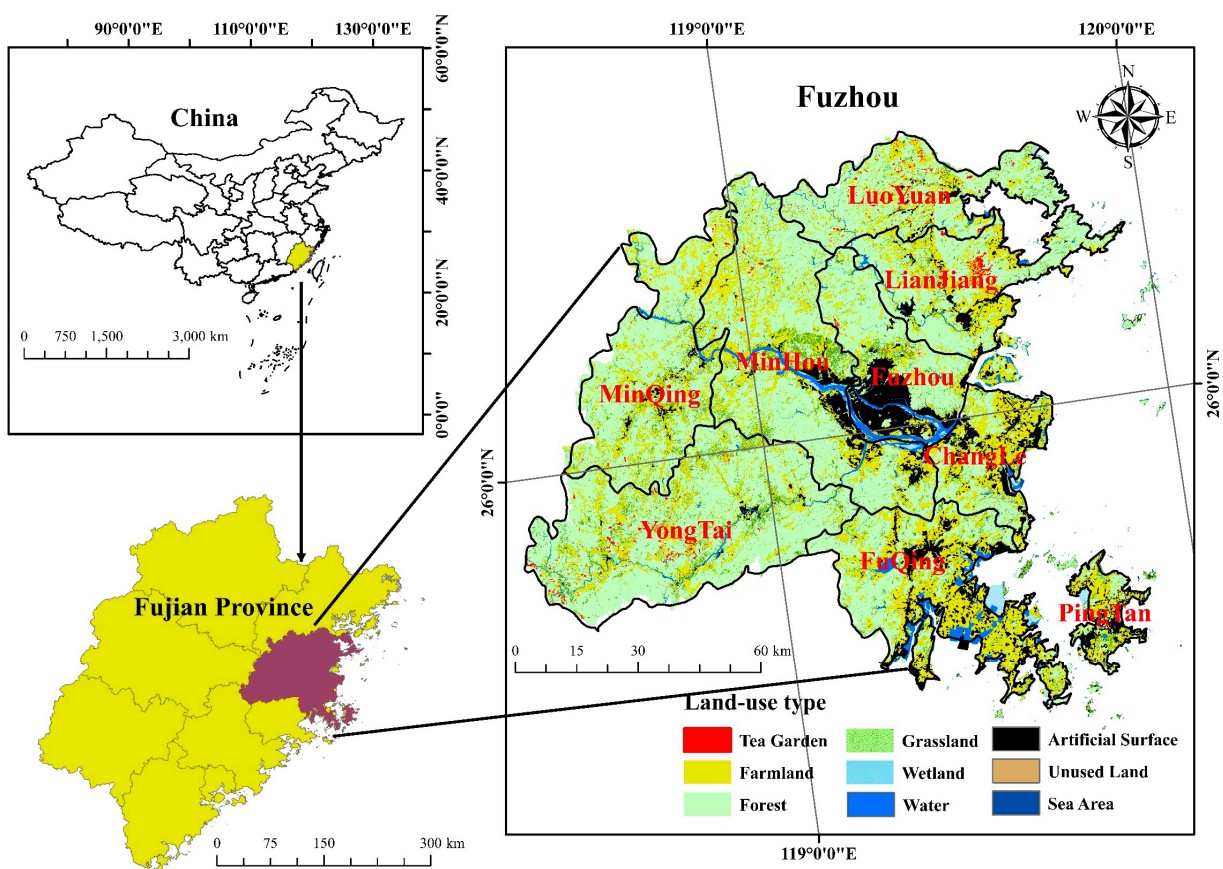

**Figure 1.** Schematic diagram of the study area.

### 2.2. Data Source

The main data selected in this study, as detailed in Table 1, were as follows:

Basic geographic data: counties, rivers, roads and the digital elevation model (DEM) data of Fuzhou, downloaded from a geospatial data cloud.

Remote sensing data: land use/land cover (LULC) was downloaded from the Institute of Remote Sensing and Digital Earth, Chinese Academy of Sciences. The soil data were downloaded from the Harmonized World Soil Database Version 1.2 (HWSD), which includes soil type, soil texture, soil organic carbon content and root depth, with a scale of 1:1 million. Vegetation remote sensing data came from MODIS data products released by the National Aeronautics and Space Center (NASA).

Meteorological data: the temperature, precipitation and water surface evaporation data in the study area were derived from the China Meteorological Science Data Sharing Service Network.

Statistics data: social and economic statistics came from the Fuzhou Statistical Yearbook and the China Energy Statistical Yearbook.

All spatial data were unified into the WGS_1984_Albers projection with a spatial resolution of 30 m × 30 m for both 2010 and 2020.

**Table 1.** The main data and data sources. Tea: tea production; WY: water yield; CR: climate regulation; SR: soil retention; SE: soil erosion; and CE: carbon emissions.

| ESs | Data | Year(s) | Spatial Resolution | Sources |
|---|---|---|---|---|
| Tea | Tabular data of tea yield | 2010/2020 | - | Fuzhou Agriculture and Rural Bureau (http://nyj.fuzhou.gov.cn/ (accessed on 21 April 2022)) processed by MODIS data |
| | The leaf area index (LAI) | 2010/2020 | 30 m | |
| | Remote sensing satellite data | 2020 | 100 m | Google Earth (https://www.google.com/earth/versions/ (accessed on 21 April 2022)) |
| NPP/SE/SR | Vegetation remote sensing data (MODIS data) | 2010/2020 | 250 m | the National Aeronautics and Space Center (NASA) (https://ladsweb.modaps.eosdis.nasa.gov/ (accessed on 10 March 2022)) |
| | NDVI | 2010/2020 | 250 m | |
| WY | Evapotranspiration data (MOD16) | 2010/2020 | 500 m | |
| WY | Watersheds | - | - | Processed by DelineateIt of InVEST |
| NPP | Vegetation type map | - | - | Map of vegetation types in China (http://www.gisrs.cn/ (accessed on 21 March 2022)) |
| CR/SE/SR/CE/WY | Land use/land cover (LULC) | 2010/2020 | 30 m | the Institute of Remote Sensing and Digital Earth, Chinese Academy of Sciences (http://www.ceode.cas.cn/sjyhfw/ (accessed on 27 February 2022)) |
| SE/SR/WY | Soil Data | - | 1 km | the Harmonized World Soil Database Version 1.2 (HWSD) (https://www.fao.org/soils-portal/soil-survey/soil-maps-and-databases/harmonized-world-soil-database-v12/en/ (accessed on 25 October 2021)) |
| NPP | Monthly average temperature | 2010/2020 | 30 m | the China Meteorological Science Data Sharing Service Network (http://data.cma.gov.cn/ (accessed on 7 March 2022)) |
| | Monthly total solar radiation | 2010/2020 | 30 m | |
| WY/SE/SR | Annual precipitation data | 2010/2020 | 30 m | |
| NPP/SE/SR | Monthly precipitation | 2010/2020 | 30 m | |
| CR | Water surface evaporation | 2010/2020 | - | |
| | Daily temperature | 2010/2020 | - | |
| CE | Fossil fuel consumption | 2010/2020 | - | Fuzhou Statistical Yearbook (https://data.cnki.net/area/yearbook/single/N2021120210?dcode=D13 (accessed on 25 May 2022)) |
| | Population | 2010/2020 | | |
| | Convert into Standard Coal | 2010/2020 | - | China Energy Statistics Yearbook (https://data.cnki.net/Trade/yearbook/single/N2021050066?zcode=Z023 (accessed on 13 June 2022)) |
| | Carbon Emission Coefficient | - | - | IPCC Guidelines for National Greenhouse Gas Inventory (https://www.ipcc-nggip.iges.or.jp/public/2006gl/chinese/index.html (accessed on 13 June 2022)) |

*2.3. Research Methods*

In this study, ESs were quantified through models of InVEST and IUEMS. The Pearson correlation were used to find out the trade-off/synergy relationship between ESs. Finally, the sum of trade-off/synergy coefficient of ESs ($C_{ts}$)are defined to determine the overall trade-off/synergy of a region. The research framework of this study is shown in Figure 2.

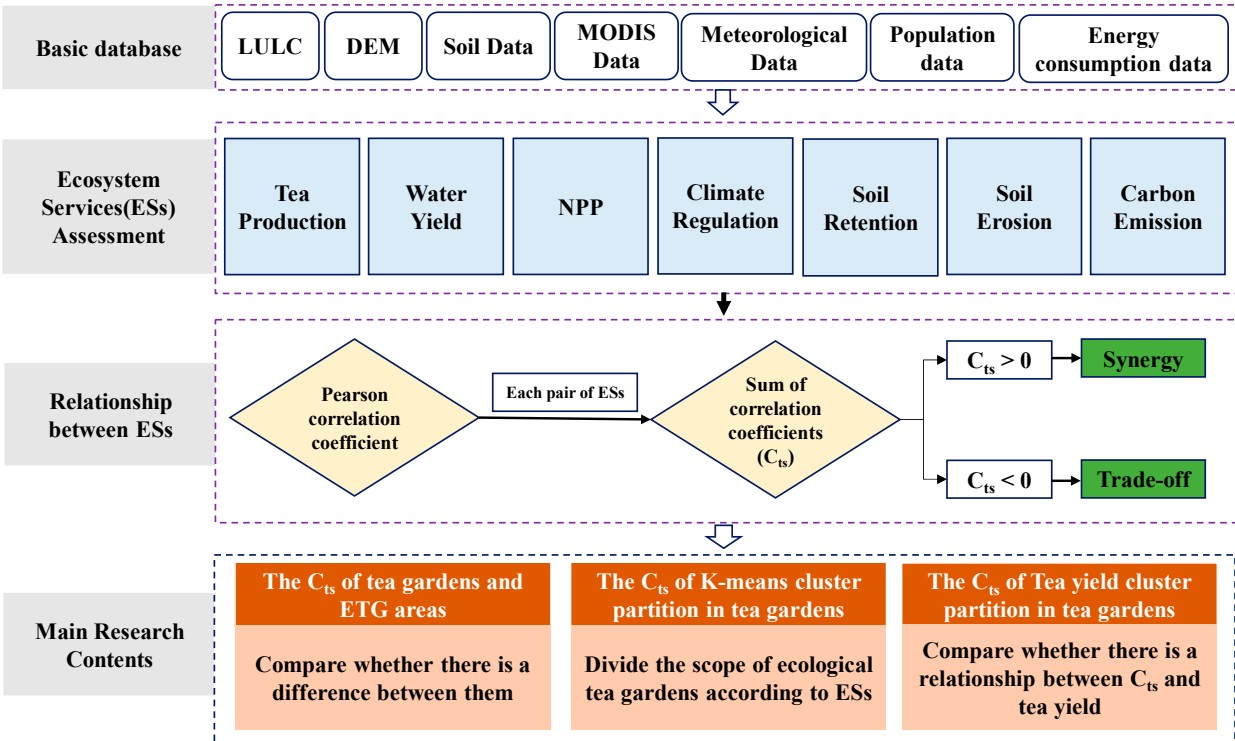

**Figure 2.** The research framework of this study. Blue represents ESs; yellow represents the research methods of relationship between ESs; green represents the relationship between ESs; and orange represents the main research contents of this study.

2.3.1. Ecosystem Service Assessment Model

(1) Integrated Valuation of Ecosystem Services and Trade-offs (InVEST)

Water yield, soil retention and soil erosion in this study were calculated based on the InVEST model, which was jointly developed by Stanford University, the Nature Conservancy (TNC) and the World-Wide Fund for Nature (WWF) (https://naturalcapitalproject. stanford.edu/software/invest (accessed on 23 March 2022). The purpose of this model is to realize the spatialization of the quantitative evaluation of the value of ecosystem service function by simulating the changes of the quality and value of the ecosystem service system under different land cover scenarios.

(2) Intelligent Urban Ecosystem Management System (IUEMS)

Climate regulation in this study is calculated based on the IUEMS model, which is a software platform for urban ecosystem assessment and planning and management led by the Center for Ecological Environment Research, Chinese Academy of Sciences, Beijing, China (https://www.iuems.com/eco/index.html (accessed on 13 April 2022). This model is operated online, with the standardized process and reliable results, and easy to operate.

2.3.2. Ecosystem Service Assessment

Based on the classification of ESs and the actual condition in Fuzhou, seven typical ESs were selected: (i) supply services: tea production and water yield; (ii) regulation services: soil retention, NPP and climate regulation; and (iii) ecosystem disservices: soil erosion and carbon emissions. The evaluation methods of each ecosystem service are as follows:

(1) Tea Production

It is proved that the leaf area index (LAI) has a good correlation with crop yield, and it is often used to characterize and predict crop yield [42,43]. Therefore, the LAI and tea production data of each county provided by the authoritative statistics department were used to draw the tea yield map of Fuzhou. In this study, tea production mainly refers to the ability of tea gardens to produce fresh tea leaves. First, we interpreted a remote sensing image of Fuzhou in 2020, acquired by Google Earth, and identified the distribution of tea gardens. Since the tea growing area is stable, it is assumed not change much within 10 years. Thus, this study took the interpreted tea gardens in 2020 as the tea gardens distribution for both 2010 and 2020, based on which tea production was calculated. Then, to map the tea production based on LAI, the tea planting distribution and tea yield of each county, we first clipped LAI according to each county of Fuzhou City. Second, the LAI attribute table of each county was exported and the sum of the LAI of each county was calculated. Finally, tea production in each pixel was calculated with Formula (1) for each county separately and then united together. The calculation method is shown in the formula:

$$Y_{tea} = \frac{i_{LAI}}{I_{LAI}} \times Y_{TEA} \tag{1}$$

where $Y_{tea}$ is the tea yield corresponding to each grid (30 m $\times$ 30 m), $i_{LAI}$ is the LAI of each grid, $I_{LAI}$ is the total LAI corresponding to Fuzhou tea gardens and $Y_{TEA}$ is the total annual tea yield of Fuzhou.

(2) Water Yield

Water yield (WY) refers to the surface water yield in a certain area [44]. The calculation of surface water yield is based on a simplified hydrological cycle model that ignores the influence of groundwater and is determined by many parameters, such as rainfall, evapotranspiration, soil depth and available water for plants. The water supply service is evaluated by the "Water Yield" module of InVEST3.12.0, and the calculation method is shown in Formula (2):

$$Y(x) = \left(1 - \frac{AET(x)}{p(x)}\right) \times p(x) \tag{2}$$

where $Y(x)$ is the water yield of land-use grid $x$, $AET(x)$ is the annual actual evapotranspiration of grid units and $p(x)$ is the annual precipitation of grid unit $x$.

In the water balance formula, the hypothesis formula of the Budyko hydrothermal coupling balance proposed by Fuh [45] and Zhang [46] is adopted to calculate the evapotranspiration of vegetation of land use/cover type $\frac{AET(x)}{p(x)}$:

$$\frac{AET(x)}{P(x)} = 1 + \frac{PET(x)}{P(x)} - \left[1 + \left(\frac{PET(x)}{P(x)}\right)^{\omega}\right]^{\frac{1}{\omega}} \tag{3}$$

where $PET(x)$ is the potential evapotranspiration and $\omega$ is a non-physical parameter that characterizes the natural climatic-soil properties.

$PET(x)$ is defined as:

$$PET(x) = K_c(l_x) \times ET_0(x) \tag{4}$$

where, $ET_0(x)$ is the reference evapotranspiration from grid unit $x$ and $K_c(l_x)$ is the plant (vegetation) evapotranspiration coefficient associated with the LULC on grid unit $x$ (Table 2).

Using the expression proposed by Donohue [47] in the InVEST model, $\omega$ is defined as:

$$\omega(x) = Z\frac{AWC(x)}{P(x)} + 1.25 \tag{5}$$

where, $AWC(x)$ is the volumetric (mm) plant available water content. It is estimated as the product of the plant available water capacity (*PAWC*) and the minimum of root restricting layer depth and vegetation rooting depth:

$$AWC(x) = Min(layer\ depth,\ root\ depth) \times PAWC \tag{6}$$

**Table 2.** Biophysical table using in water yield.

| Land-Use Type | Coefficient of Evapotranspiration | Plant Root Depth (mm) | Category |
|---|---|---|---|
| Farmland | 0.8 | 1000 | 1 |
| Forest | 1 | 7000 | 1 |
| Grassland | 0.7 | 1700 | 1 |
| Wetland | 1 | 1 | 0 |
| Water Body | 1.2 | 1 | 0 |
| Artificial Surface | 0.3 | 1 | 0 |
| Unused Land | 0.5 | 1 | 0 |
| Sea Area | 1.3 | 1 | 0 |

Root restricting layer depth is the soil depth at which root penetration is inhibited because of physical or chemical characteristics. Vegetation rooting depth is often given as the depth at which 95% of a vegetation type's root biomass occurs. *PAWC* is the plant available water capacity. *Z* is an empirical constant, which captures the local precipitation pattern and additional hydrogeological characteristics and is positively correlated with *N*, the number of rain events per year. According to previous studies, the *Z* value is determined as 11.5 [48]. The 1.25 term is the minimum value of $\omega(x)$, which is the value for bare soil (when root depth is 0).

The evapotranspiration coefficient and plant root depth parameters were set according to the InVEST model guidance manual, FAO's Handbook for Calculating Crop Evapotranspiration—Crop Water Demand and previous studies [48,49]. The category is defined as 1 or 0, depending on whether there is vegetation cover. When the vegetation cover is 0, the model will skip the index of plant root depth, so it is set as 1.

(3) Soil Retention and Soil Erosion

Soil erosion (SE) is the displacement of the upper layer of soil, one form of soil degradation. This natural process is caused by the dynamic activity of erosive agents, that is, water, ice, snow, air(wind), plants, animals, and humans [36]. The soil retention (SR) function represents when ecosystems (such as farmland, forest, etc.) reduce the erosion energy of rainwater through the canopy, litter, roots, and other levels and increase soil erosion resistance to reduce soil erosion, reduce soil loss and maintain soil [50]. This parameter is evaluated by the sediment transport ratio (SDR) module of InVEST3.12.0, which is mainly based on the universal soil loss equation (USLE), and uses various datasets, such as topography, climate, vegetation and management practices to calculate the annual average soil loss and soil retention of each land-type grid. The calculation method is shown in Formula (7):

Calculation of potential soil erosion based on geomorphology and climatic conditions:

$$RKLS = R \times K \times LS \tag{7}$$

Actual soil erosion considers vegetation cover and soil and water conservation measures, which represent the soil erosion selected for this study:

$$USLE = R \times K \times LS \times C \times P \tag{8}$$

Soil retention is the difference between *RKLS* and *ULSE*:

$$A_{SR} = RKLS - ULSE \tag{9}$$

where $A_{SR}$ represents the soil retention, $RKLS$ represents the potential soil erosion, $USLE$ represents the actual soil erosion, $R$ represents the rainfall erosivity, $K$ represents the soil erodibility factor, $LS$ represents the slope length factor, $C$ represents the vegetation cover factor and management factor and $P$ represents the soil and water conservation measure factor.

The $R$ factor is generally calculated by the Wischmeier [51] formula based on monthly average precipitation and annual average precipitation:

$$R = \sum_{i=1}^{12} 1.735 \times 10^{(1.5 \cdot lg \frac{p_i^2}{p}) - 0.8188} \tag{10}$$

where $p_i$ is the average monthly precipitation (mm) and $p$ is the average annual precipitation (mm).

The $K$ factor is calculated mainly through the EPIC model [52], and the formula is as follows:

$$K_{USLE} = K = f_{csand} \times f_{cl-si} \times f_{orgc} \times f_{hisand} \tag{11}$$

$$f_{csand} = \left(0.2 + 0.3 \times exp\left[-0.256 \times m_s \times \left(1 - \frac{m_{silt}}{100}\right)\right]\right) \tag{12}$$

$$f_{cl-si} = \left(\frac{m_{silt}}{m_c + m_{silt}}\right)^{0.3} \tag{13}$$

$$f_{orgc} = \left(1 - \frac{0.0256 \times orgc}{orgc + exp[3.72 - 2.95 \times orgc]}\right) \tag{14}$$

$$f_{hisand} = \left(1 - \frac{0.7 \times \left(1 - \frac{m_s}{100}\right)}{\left(1 - \frac{m_s}{100}\right) + exp\left[-5.51 + 22.9 \times \left(1 - \frac{m_s}{100}\right)\right]}\right) \tag{15}$$

where $m_s$ is the content of sand grain (%), $m_{silt}$ is the content of silty sand (%), $m_c$ is the content of clay particles (%) and $orgc$ is the content of organic carbon (%).

The LS factor includes two aspects: slope length factor $L$ and slope factor $S$.

The calculation of the $L$ factor refers to the formula proposed by Wischmeier and Smith [53], and the specific calculation formula is as follows:

$$L = \left(\frac{\gamma}{22.13}\right)^{\alpha} \tag{16}$$

$$\alpha = \frac{\beta}{\beta + 1} \tag{17}$$

$$\beta = \left(\frac{\sin\theta}{0.0896}\right) / \left[3.0(\sin\theta)^{0.8} + 0.56\right] \tag{18}$$

where $\gamma$ is the length of the horizontal slope, $\alpha$ is the index of slope length and $\theta$ is the slope.

The calculation of the $S$ factor refers to the formula proposed by McCool [54], and the specific calculation formula is as follows:

$$\begin{cases} S = 10.8 \times \sin\theta + 0.03 \ (\theta < 5°) \\ S = 16.8 \times \sin\theta - 0.50 \ (5° \leq \theta \leq 10°) \\ S = 21.91 \times \sin\theta - 0.96 \ (\theta > 10°) \end{cases} \tag{19}$$

where $\theta$ is the slope.

The calculation of the $C$ factor is mainly based on $NDVI$ data, and the specific calculation formula is as follows [55]:

$$b = \frac{NDVI - NDVI_{min}}{NDVI_{max} - NDVI_{min}} \tag{20}$$

$$C = \begin{cases} 1 & b = 0 \\ 0.6508 - 0.3436 \ln b & 0 < b \leq 78.3\% \\ 0 & b > 78.3\% \end{cases} \tag{21}$$

where $b$ is the vegetation coverage.

The determination of the P factor (Equation (8)) is mainly based on previous studies [56,57]. The P factor values of different land-use types are listed in Table 3.

**Table 3.** Soil and water conservation measure factors (P).

| Land-Use Type | P |
|---|---|
| Farmland | 0.45 |
| Forest | 0.4 |
| Grassland | 0.7 |
| Wetland | 0 |
| Water Body | 0 |
| Artificial Surface | 1 |
| Unused Land | 1 |
| Sea Area | 0 |

(4) NPP

NPP represents the carbon sequestration and oxygen release function by which natural ecosystems absorb $CO_2$ in the atmosphere to synthesize organic matter during photosynthesis, sequester carbon in plants or soil and release oxygen [58].

The improved CASA model was used to calculate the NPP of vegetation [59], and the calculation method is shown in the formula:

$$NPP(x,t) = APAR(x,t) \times \varepsilon(x,t) \tag{22}$$

$$APAR(x,t) = SOL(x,t) \times FPAR(x,t) \times 0.5 \tag{23}$$

$$\varepsilon(x,t) = T_{\varepsilon1}(x,t) \times T_{\varepsilon2}(x,t) \times W_{\varepsilon}(x,t) \times \varepsilon_{max} \tag{24}$$

where $NPP(x,t)$ is the total organic matter accumulation of plants in month $t$ at pixel $x$ [g C/(m$^2$ · month)], $APAR(x,t)$ is the effective photosynthetic radiation absorbed in month $t$ at pixel $x$ [MJ/(m$^2$ · month)], $\varepsilon(x,t)$ is the actual light energy utilization of plants in month $t$ at pixel $x$, $SOL(x,t)$ is the total radiation of the sun in month $t$ at pixel $x$ [MJ/(m$^2$ · month)], $FPAR(x,t)$ is the ratio of effective photosynthetic radiation absorbed by plants in month $t$ at pixel $x$, 0.5 is the ratio of effective photosynthetic radiation to total solar radiation, $T_{\varepsilon1}(x,t)$ and $T_{\varepsilon2}(x,t)$ are the influence coefficients of low temperature and high temperature stress, $W_{\varepsilon}(x,t)$ is the influence coefficient of water stress and $\varepsilon_{max}$ is the maximum utilization rate of light energy under ideal conditions (%).

(5) Climate Regulation

Climate regulation (CR) is the function by which natural ecosystems absorb solar energy through the transpiration of vegetation and evaporation of the water surface to regulate the summer temperature and improve the suitability of human settlements [60]. The climate regulation service in this study is calculated by IUEMS. Taking the total energy consumed by ecosystem transpiration and evaporation as the function quantity for regulating climate, the calculation method is shown in the formula:

$$E_{tt} = E_{pt} + E_{we} \tag{25}$$

$$E_{pt} = \sum_{i}^{3} EPP_i \times S_i \times D \times 10^6 / (3600 * r) \tag{26}$$

$$E_{we} = E_w \times q \times 10^3 / (3600) \tag{27}$$

where $E_{tt}$ is the total energy consumed by ecosystem transpiration (kWh/a), $E_{pt}$ is the energy consumed by ecosystem vegetation transpiration (kWh/a), $E_{we}$ is the capacity consumed by ecosystem water surface evaporation (kWh/a), $EPP_i$ is the heat consumed by

transpiration per unit area of the class $i$ ecosystem (KJ·m$^{-2}$·d$^{-1}$), $D$ is the number of days in a year when the highest temperature is greater than 26°C, $r$ is a dimensionless number equal to 3.0 (dimensionless), $i$ is the land-use type of ecosystem ($i$ = 1,2,3 ... 8) (farmland, forests, grasslands, wetland, water body, sea areas, artificial surfaces and unused land), $E_w$ is the water surface evaporation (m$^3$) and $q$ is the latent heat of evaporation, that is, the heat required to evaporate 1 g of water (J/g).

(6) Carbon Emissions

The carbon emission (CE) disservice characterized by land-use type is the difference between the carbon source and carbon sink [61]. In this study, artificial surfaces, cultivated land and tea gardens regarded as carbon sources, forests, grasslands, water bodies, sea areas and unused land are identified as carbon sinks. In this study, the LULC carbon emission calculation method was used to calculate carbon emissions, which can be divide into direct carbon emissions and indirect carbon emissions [62]. The carbon emissions (carbon absorption) of cultivated land, tea gardens, forests, grasslands, wetland, water body, sea areas and unused land were calculated by the direct carbon emission coefficient method:

$$C_i = \sum A_i \times \alpha_i \qquad (28)$$

where $C_i$ is the carbon emissions (carbon absorption) of the $i$ land-use type ($i$ = 1, 2, 3 ... 8) (farmland, tea gardens, forests, grasslands, wetland, water body, sea areas and unused land), $A_i$ is the area of the $i$ land-use type and $\alpha_i$ is the carbon emission (carbon absorption) coefficient of the $i$ land-use type.

Based on previous studies and the actual characteristics of Fuzhou, the carbon emission coefficient of each land-use type was determined from the literature and is presented in Table 4 [63,64].

**Table 4.** Land-use type carbon emission factor. (Emissions are positive, and absorptions are negative.)

| Land-Use Type | Farmland | Forest | Grassland | Wetland | Water Body | Unused Land | Sea Area | Tea Garden |
|---|---|---|---|---|---|---|---|---|
| CE coefficient (t/hm$^2$) | 0.4970 | −5.1100 | −0.9490 | −0.4100 | −0.2520 | −0.0050 | −0.2510 | 0.4220 |

Artificial surface refers to the type of surface cover formed by human activities and is covered by asphalt, concrete, sand, brick, glass and other building materials, including urban and other residential areas, industrial and mining facilities and transportation facilities [65]. Artificial surfaces represent a large number of activities of human energy consumption, and we assume in this study that carbon emissions from fossil energy consumption are concentrated on artificial surfaces. Artificial surfaces' carbon emissions are estimated indirectly, mainly to calculate the carbon emissions generated when coal, coke, crude oil, gasoline, kerosene, diesel oil, fuel oil, natural gas and electricity are consumed in the construction process. The calculation formula is as follows:

$$C_i = \sum m_i \times n_i \times \varphi_i \qquad (29)$$

In this formula, $C_i$ is the carbon emissions of the artificial surface, $m_i$ is the consumption of the $i$ energy, $n_i$ is the standard coal conversion coefficient of the $i$ energy, and $\varphi_i$ is the carbon emission coefficient of the $i$ energy. The fossil energy consumption and population data were mainly obtained from the Fuzhou Statistical Yearbook. The conversion coefficient of fossil energy into standard coal and the carbon emission coefficient were mainly derived from the China Energy Statistics Yearbook and IPCC Guidelines for National Greenhouse Gas Inventory [66]. The carbon emission coefficient of each fossil energy source is listed in Table 5.

**Table 5.** Fossil energy carbon emission factors.

| Fossil Fuel Types | Convert Into Standard Coal | Carbon Emission Coefficient |
|:---:|:---:|:---:|
| Raw Coal | 0.7143 | 0.7559 |
| Coke | 0.9714 | 0.855 |
| Petrol | 1.4714 | 0.5338 |
| Paraffin | 1.4714 | 0.5714 |
| Diesel | 1.5714 | 0.5912 |
| Fuel Oil | 1.4286 | 0.6185 |
| LPG | 1.7143 | 0.5042 |
| Natural Gas | 1.2143 | 0.4483 |
| Power | 0.1229 | 2.5255 |

2.3.3. Correlation Analysis and Trade-Off/Synergy Coefficients of ESs

Since the tea garden areas account for a small proportion of Fuzhou, the scale is quite different from that of the ETG areas; therefore, the tea garden area in this study was based on 30 m × 30 m grid points, and the ETG areas were based on 60 m × 60 m grid points. The ESs were extracted to each point, and SPSS software was used to analyze the correlation.

(1) The Pearson correlation coefficient method was used to analyze the trade-offs/synergies among ESs. The Pearson correlation coefficient is a method used to measure the degree of correlation between two variables and has been widely used to analyze the correlation analysis of two time series [67]. The correlation coefficient r is estimated by the following formula:

$$r = \frac{\sum_{i=1}^{n} \frac{(X-\overline{X})(Y-\overline{Y})}{n-1}}{\sqrt{\sum_{i=1}^{n} \frac{(X-\overline{X})^2}{n-1}} \sqrt{\sum_{i=1}^{n} \frac{(Y-\overline{Y})^2}{n-1}}} \tag{30}$$

where $n$ is the length of data, $X$ and $Y$ refer to different ecosystem services and $i$ is the current order of the ESs ($i = 1, 2, \ldots \ldots n$).

(2) There is a correlation coefficient between each pair of ESs, and we defined a total ESs trade-off/synergy correlation coefficient ($C_{ts}$), which indicates the degree of network connection between multiple ESs. $C_{ts}$ was adopted to measure the overall relationship among ESs in the region. The calculation method is shown in the formula:

$$C_{ts} = \sum_{i}^{n} r \tag{31}$$

where $r$ is the correlation coefficient between each pair of ESs, $i$ is the $i$ pair of ESs ($I = 1, 2, 3 \ldots$) and $n$ is the total logarithm of ESs.

2.3.4. Spatial Clustering Analysis

Cluster analysis classifies a batch of sample data according to different characteristics and the degree of correlation in quality [68]. In this study, the K-means clustering method was used to reclassify the ESs types. K-means clustering is a nonhierarchical clustering method that is widely used in ecological land surface classification [69]. Based on the data points of Fuzhou fishing nets, the datasets of ESs were extracted, the optimal K was calculated using the R software and cluster analysis and spatial visualization were realized in SPSS and ArcGIS software.

The method first determines the number of categories to be clustered. Then, the computer preliminarily determines the original center points of each type according to the center of the data structure, calculates the distance from each record to these center points in succession, divides it into k types according to the principle of the closest distance, recalculates the center points of each type and repeats the above process according to the new center position until the preset iteration times are reached. The formula is as follows:

K centroids are randomly selected from n sample data as the initial clustering centers. The centroid is recorded as:

$$\mu_1^{(0)}, \mu_2^{(0)}, \ldots, \mu_k^{(0)}$$

Optimization objectives are defined as:

$$J(c, \mu) = min \sum_{i=1}^{M} ||x_i - \mu_{c_i}||^2 \tag{32}$$

The loop is started and the distance from each sample point to that of the centroid is calculated. The sample to which the centroid is closest to is assigned and k clusters are obtained:

$$C_i^t < -\arg min ||x_i - \mu_k^t||^2 \tag{33}$$

For each cluster, the average distance of all sample points is divided into the cluster as the new centroid is calculated:

$$\mu_k^{(t+1)} < -argmin \sum_{i:c_i^t=k}^{b} ||x_i - \mu||^2 \tag{34}$$

Until *j* converges, all clusters do not change.

(1) Determining the number of clusters (k)

In this study, the elbow rule was adopted to determine the optimal K value by finding the inflection point where the loss value decreases smoothly. The cluster evaluation index used by elbow SSE (sum of squares of errors) is the square of the sum of distances from all sample points in the dataset to their cluster centers, and the formula is as follows:

$$SSE = \sum_{i=1}^{K} \sum_{p \in C_i} |p - m_i|^2 \tag{35}$$

where $C_i$ is the *i* cluster; *p* is the sample points in $C_i$, $m_i$ is the mean of all samples in $C_i$ and *SSE* is the clustering error of all samples, which represents the clustering effect.

$m_i$ is the centroid of the i cluster.

Based on the above formula, the optimal k is calculated in the R software, and finally, the cluster number k = 5 is selected to form the elbow map and is shown in Figure 3.

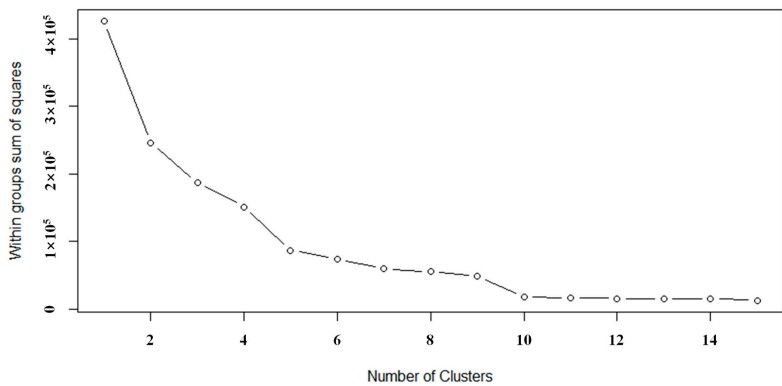

**Figure 3.** Diagram of the elbow rule for determining K-means. The y-axis is the SSE and the x-axis is the value of k. As x increases, the SSE decreases, and when the decrease tends to be significantly slower, the value is considered to be the best value of k.

(2) Determining the initial clustering center

The initial cluster center is randomly selected by the SPSS software, and the final cluster center is obtained after iterative calculation.

(3) Setting the iteration threshold and calculation

After the iteration threshold is set to 100, the calculation begins. After 97 iterations, the final calculation results converge. The final clustering results and the clustering information of each cluster member are obtained.

(4) Obtaining the final clustering center

## 3. Results

### 3.1. Tea Production and ESs Mapping in Fuzhou

The spatial distribution patterns of ESs in Fuzhou in 2010 and 2020 are shown in Figure 4. The distribution of tea plantations in Fuzhou was relatively scattered (Figure 4a), mainly located in Yongtai County, Lianjiang County and Luoyuan County. After calculating the tea yield of each county, we divided the relatively high-value area and relatively low-value area of tea yield. The high-value areas of tea production in 2010 and 2020 were mainly distributed in Lianjiang County, with yields up to 3.6 t/ha. The low-value areas in Yongtai County yielded 0.1–1.0 t/ha. The distribution of WY shows spatial differences (Figure 4b), with high-value areas mainly in the western region with higher vegetation cover and low values in the eastern region.

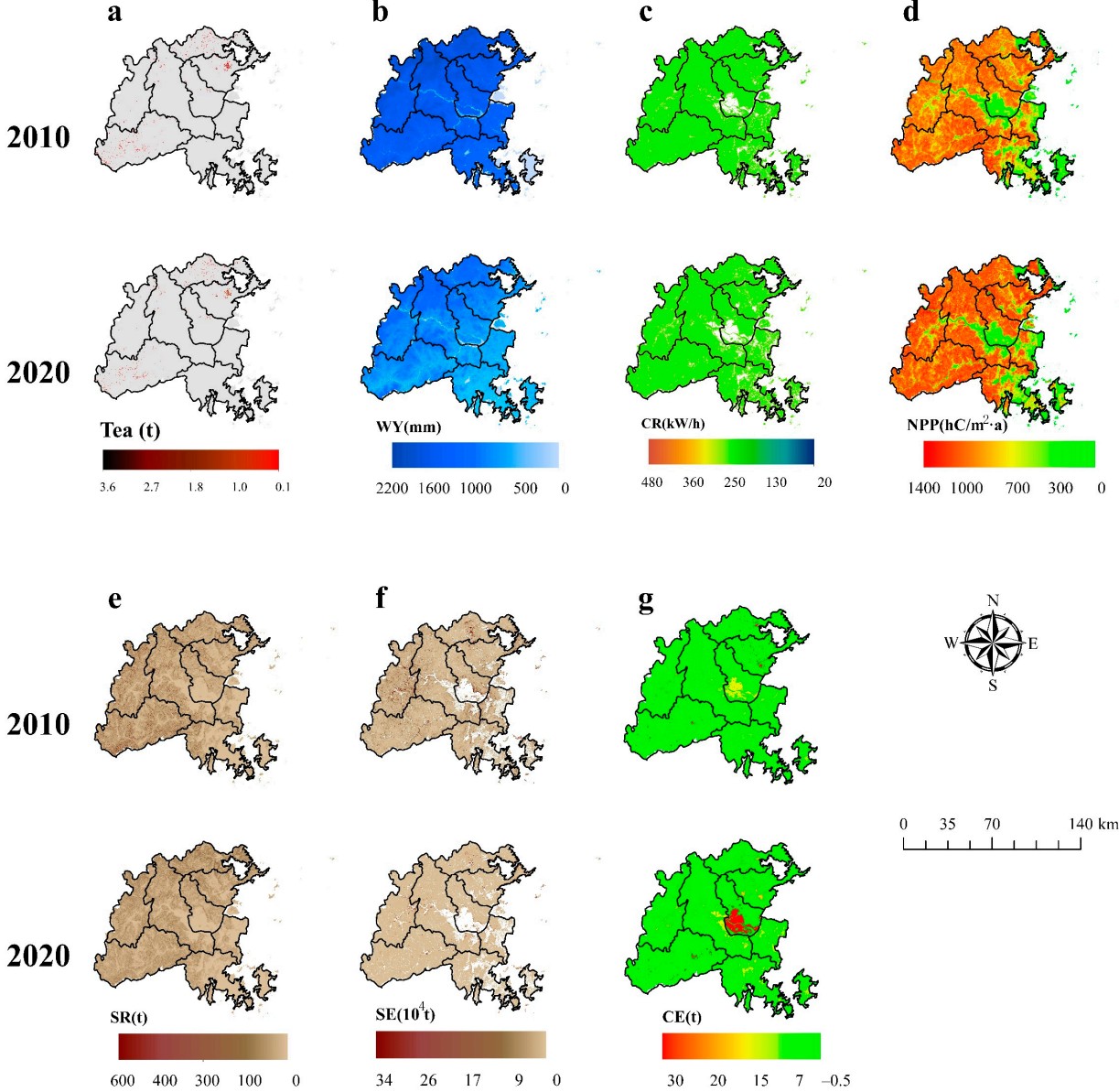

**Figure 4.** ES patterns in 2010 and 2020 of Fuzhou city, China. Tea: tea production; WY: water yield; CR: climate regulation; SR: soil retention; SE: soil erosion; and CE: carbon emissions. (**a**): tea production in Fuzhou in 2010 and 2020; (**b**): water yield in Fuzhou in 2010 and 2020; (**c**): climate regulation in Fuzhou in 2010 and 2020; (**d**): NPP in Fuzhou in 2010 and 2020; (**e**): soil retention in Fuzhou in 2010 and 2020; (**f**): soil erosion in Fuzhou in 2010 and 2020; (**g**): carbon emissions in Fuzhou in 2010 and 2020.

The spatial distribution of CR is shown in Figure 4c. The high-value areas in 2010 and 2020 were mainly distributed in urban areas and the eastern seaboard. The low-value areas were mostly distributed in areas with less human activities in the study area. NPP and SR show the similar spatial distribution pattern (Figure 4d,e). The high-value areas in 2010 and 2020 were mostly distributed in areas with higher vegetation cover and higher elevation in the northwestern and southeastern part of the study area, while the low-value areas were mainly distributed in urban areas.

The spatial distribution of SE is shown in Figure 4f. The high-value areas in 2010 and 2020 were mostly distributed on the western edge of the study area, and the low-value areas were mainly distributed in urban areas and the eastern coast. The spatial distribution of CE is shown in Figure 4g. The high-value areas in 2010 and 2020 were mainly distributed in urban areas, while other areas were in a lower carbon emission state or carbon absorption state.

### 3.2. Correlation Analysis of ESs between Tea Gardens and ETG Areas

To interpret the gap between tea and ETG areas in the correlation of ESs, and based on Pearson correlation coefficient analysis, the correlation analysis of each pair of ESs was performed for 2010 and 2020 (Figure 5). We excluded CE in the correlation of tea gardens because of the calculation based on land-use type, which indicate the same CE from tea gardens. The correlation of each pair of ESs passed the $p < 0.01$ significance test. Tea production had a positive correlation with CR and WY services in 2010 and 2020, which suggests a synergistic relationship.

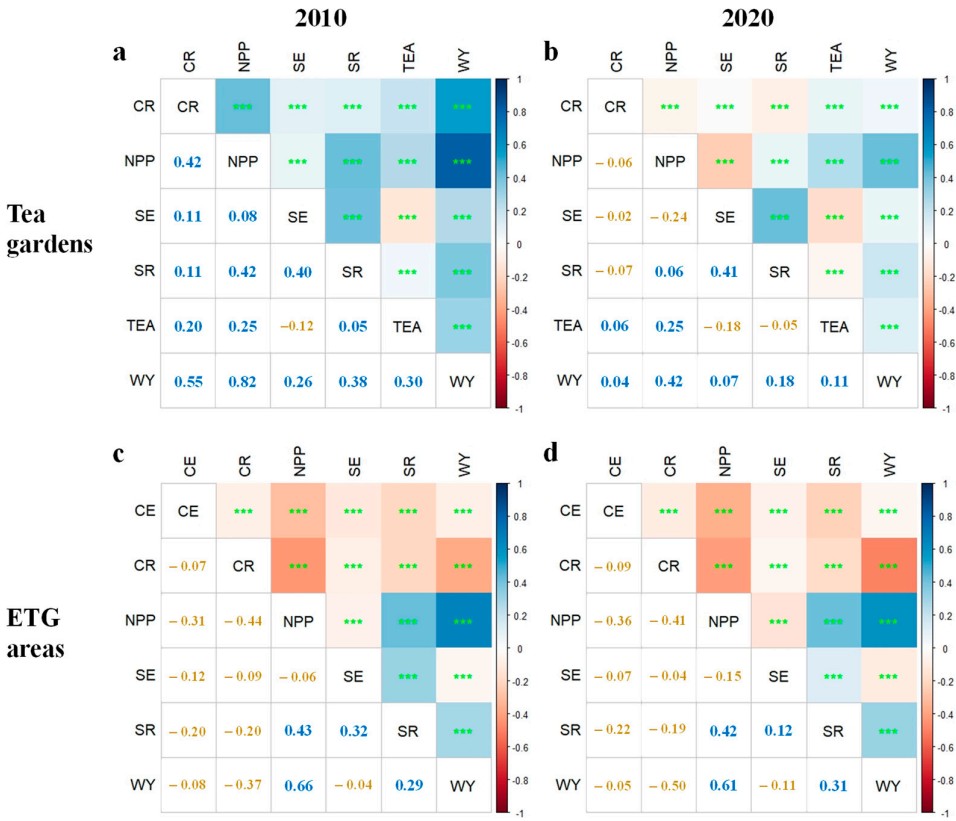

**Figure 5.** Diagram of the correlation between ESs in Fuzhou. TEA: tea production; WY: water yield; CR: climate regulation; SR: soil retention; SE: soil erosion; and CE: carbon emissions. (**a**): Pearson's correlation index between ESs in tea gardens of Fuzhou in 2010; (**b**): Pearson's correlation index between ESs in tea gardens of Fuzhou in 2020; (**c**): Pearson's correlation index between ESs in ETG areas of Fuzhou in 2010; (**d**): Pearson's correlation index between ESs in ETG areas of Fuzhou in 2020 (*** indicates significant at the 0.001 level).

There was a visual change in the correlation of ESs in tea gardens (Figure 5a,b). In 2010, there was a synergistic relationship between ESs, except for tea production and SE. In 2020, tea production shows a negative correlation with SE and SR, suggesting a trade-off relationship. The synergy between WY and other services declined to some extent. The relationship between NPP with CR and SE shifted from synergy to trade-off, also indicating a degree of ecological quality decline in tea gardens from 2010 to 2020.

In ETG areas (Figure 5c,d), there was a positive correlation among SR, WY, NPP and SE, showing a synergistic relationship, and there was a negative correlation among other ESs. The ES correlation of ETG areas remained stable with little overall change from 2010 to 2020.

To discover the difference in ES correlation between tea garden areas and ETG areas, we compared the $C_{ts}$ of all ESs in tea garden areas and ETG areas (Figure 6). The $C_{ts}$ in the tea garden area was positive, showing a synergistic relationship. Similarly, the declining $C_{ts}$ in the tea garden areas indicates the ecological quality decline of the tea garden areas in Fuzhou from 2010 to 2020. The $C_{ts}$ in ETG areas was negative, showing a trade-off relationship.

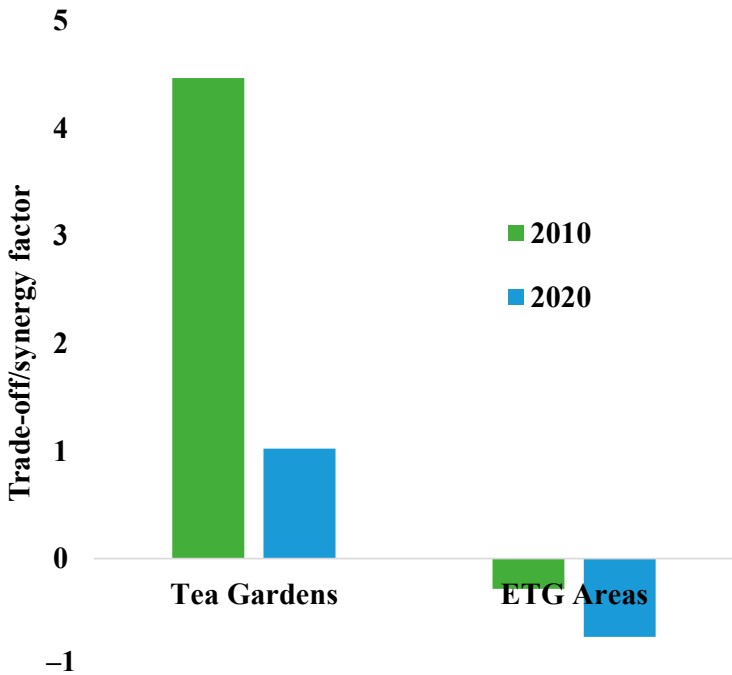

**Figure 6.** $C_{ts}$ in tea garden and ETG areas in Fuzhou.

### 3.3. Clustering Analysis of Tea Garden ESs in Fuzhou

According to the K-means clustering number determined in the research method, we calculated it in SPSS and finally determined the clustering centers (Figure 7). We excluded CE because of the calculation based on land-use type, which indicate the same CE from tea gardens. The final clustering centers represent the basis for the division of the tea gardens' ESs, and ultimately the cluster to which each sample point belongs was determined based on the shortest distance from each sample point to the cluster center. The clustering center of Area I exhibited the prominent characteristics of SE, and Area II shows a dominant WY. The clustering center of Area III exhibits a more balanced feature among ESs. The cluster center of Area IV was similar with Area III but with higher tea production and lower SE. The cluster center of Area V exhibits high tea production with low SE and SR.

Based on the ESs of the tea garden areas and final clustering centers (Figure 7), Fuzhou tea gardens in 2010 and 2020 were divided into five districts based on the K-means clustering. The final clustering results and statistics of the partition area are shown in Table 6. We found a dramatic decrease in Area I and Area IV and an increase in Area V. On the

basis of the division of ES functions, a rose diagram of each district was created (Figure 8), the ES structure in each ecosystem service function district was analyzed and the dominant ecosystem service types and the trade-offs and synergies among ESs in each district were identified.

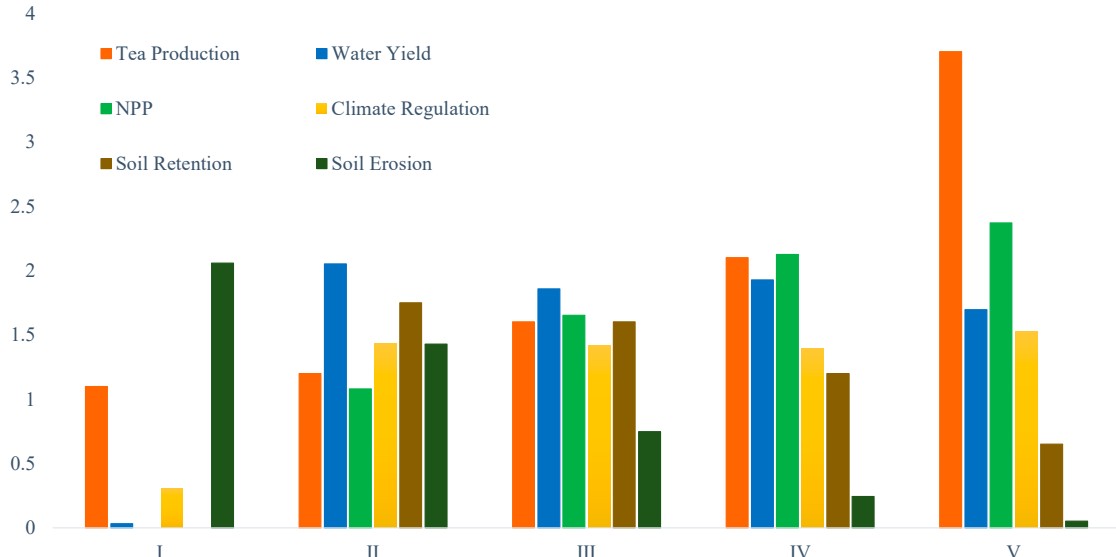

**Figure 7.** Final clustering centers for K-means clustering of the tea gardens. To show the relationship of the ESs, we scaled the ESs in the same dimension so that they can be displayed in the same field of view.

**Table 6.** Statistics of the tea garden ES partitions in Fuzhou.

| Partition | Area in 2010 (ha) | Proportion in 2010 (%) | Area in 2020 (ha) | Proportion in 2020 (%) |
|---|---|---|---|---|
| I | 1074.96 | 14.01 | 164.43 | 2.14 |
| II | 39.51 | 0.52 | 8.64 | 0.11 |
| III | 240.03 | 3.13 | 129.33 | 1.69 |
| IV | 2027.97 | 26.45 | 742.14 | 9.68 |
| V | 4285.44 | 55.89 | 6623.37 | 86.38 |

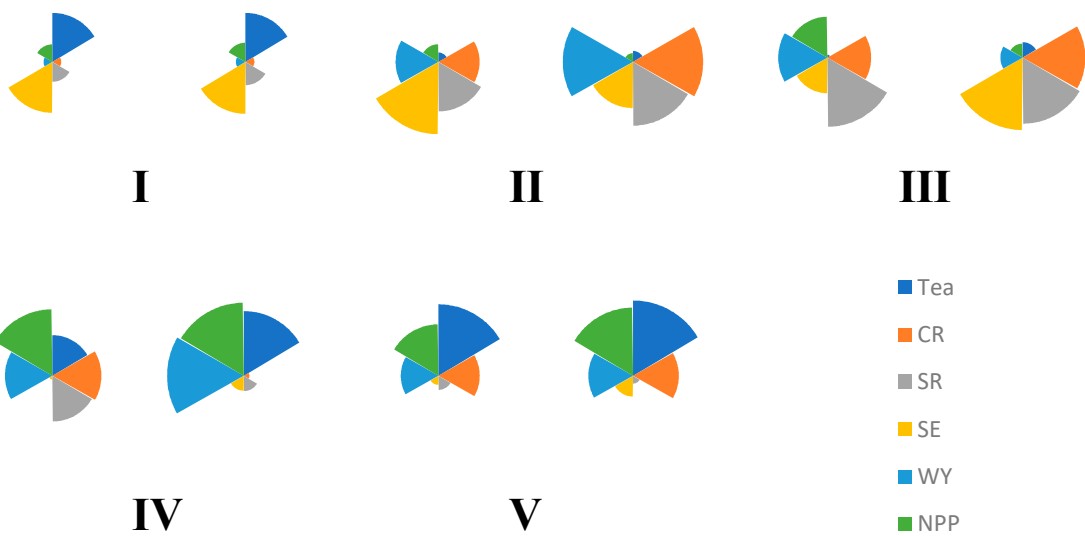

**Figure 8.** Fuzhou tea garden regional ecosystem service cluster (left: 2010; right: 2020). Tea: tea production; WY: water yield; CR: climate regulation; SR: soil retention; SE: soil erosion; and CE: carbon emissions.

Area I (Traditional Tea Garden): tea production and SE are relatively dominant, while CR and WY are relatively weak. Area II (Ecological Tea Garden Construction Area): SE is the most dominant; CR, SR and WY are relatively dominant; tea production is the weakest; and SE accounted for a larger proportion in Area II than in Area I. Area III (Ecological Tea Garden Buffer Area): compared with Area II, the SR is the most dominant; CR, SE, WY and NPP are relatively dominant; and SR accounted for a larger proportion in Area III than that in Area II. Area IV (Mature Ecological Tea Garden Area): Area IV has the best ecological environment and, except for SR, the proportion of ESs is relatively balanced. Area V (Immature Ecological Tea Garden Area): tea production is dominant, which is the most dominant area among the five districts, while SR and SE are relatively weak. The difference between Area IV and V is that Area V has the highest tea production, but the protection of the ecological environment must be strengthened.

### 3.4. Trade-Off/Synergy Analysis of Tea Garden Districts in Fuzhou Based on Yield

Based on the obtained tea production data of Fuzhou in 2010 and 2020, the average yield data were reclassified, and the tea garden in Fuzhou was divided into five districts. The final reclassification results and statistics of district areas are listed in Table 7. The areas with low tea production (Zone I and Zone II) accounted for 70.87% of the total tea garden area, while the areas with high tea yield (Zone V) only accounted for 0.15% of the total tea garden area.

**Table 7.** Statistics on the reclassified areas of the tea gardens in Fuzhou.

| Partition | Tea Yield (kg/m$^2$) | Area (ha) | Proportion (%) |
| --- | --- | --- | --- |
| I | 8.22–27.04 | 2342.52 | 41.78 |
| II | 27.04–38.62 | 1631.07 | 29.09 |
| III | 38.62–53.10 | 1287 | 22.95 |
| IV | 53.10–114.62 | 338.04 | 6.03 |
| V | 114.62–192.79 | 8.64 | 0.15 |

To explain the correlation between tea production and ES trade-offs/synergies, based on Pearson correlation coefficient analysis, correlation analyses of the seven ESs, WY, NPP, CR, SR, SE, CE and tea production, in the five regions in 2010 and 2020 were performed. The results are shown in Table 8, and all results passed the 0.01 significance test. The higher the tea production of the partition, the stronger the synergistic relationship between ESs in both 2010 and 2020.

Tea production in 2010 basically shows a synergistic relationship with WY and NPP and a trade-off relationship with other services. In 2020, WY and tea production exhibited a trade-off relationship. Only NPP and tea production show a synergistic relationship, and the synergistic relationship became stronger with the increase in tea production in different districts.

After adding the correlation coefficients of ESs in each zone, the $C_{ts}$ in each zone was positive, showing a synergistic relationship. With the increase in tea production, the synergistic relationship accelerated, showing the same trend in 2010 and 2020 (Figure 9). The respective change across zones is from 2.6 to 7.5 in 2010, while in 2020, the change is from 1.9 to 6.5.

**Table 8.** Correlation coefficients among ESs of tea garden yield zoning in Fuzhou. Tea: tea production; WY: water yield; CR: climate regulation; SR: soil retention; SE: soil erosion; and CE: carbon emissions. Red indicates a positive Pearson correlation coefficient between ESs; green indicates a negative Pearson correlation coefficient between ESs.

| Ecosystem | 1 | | 2 | | 3 | | 4 | | 5 | |
|---|---|---|---|---|---|---|---|---|---|---|
| Services | 2010 | 2020 | 2010 | 2020 | 2010 | 2020 | 2010 | 2020 | 2010 | 2020 |
| Tea and CR | 0.159 | −0.085 | −0.004 | 0.047 | −0.021 | 0.024 | −0.066 | −0.134 | −0.545 | −0.701 |
| Tea and SR | 0.029 | −0.145 | −0.108 | −0.195 | −0.054 | −0.059 | −0.107 | 0.006 | 0.048 | 0.250 |
| Tea and SE | −0.196 | −0.365 | −0.227 | −0.121 | −0.061 | −0.205 | −0.249 | −0.073 | −0.124 | 0.194 |
| Tea and WY | 0.218 | −0.195 | 0.189 | −0.138 | 0.128 | −0.074 | −0.009 | −0.088 | −0.024 | −0.057 |
| Tea and NPP | 0.041 | 0.271 | −0.136 | 0.093 | −0.041 | 0.167 | 0.051 | 0.033 | −0.012 | −0.161 |
| CR and SR | 0.021 | 0.073 | −0.037 | −0.012 | −0.124 | −0.094 | 0.100 | 0.081 | 0.292 | 0.169 |
| CR and SE | 0.055 | 0.100 | 0.095 | −0.023 | 0.144 | −0.076 | 0.143 | −0.101 | 0.144 | 0.195 |
| CR and WY | 0.458 | 0.313 | 0.476 | 0.204 | 0.685 | 0.131 | 0.545 | 0.533 | 0.685 | 0.663 |
| CR and NPP | 0.186 | 0.170 | 0.261 | 0.114 | 0.063 | 0.010 | 0.329 | 0.417 | 0.676 | 0.761 |
| SR and SE | 0.342 | 0.532 | 0.496 | 0.386 | 0.641 | 0.523 | 0.756 | 0.344 | 0.931 | 0.756 |
| SR and WY | 0.206 | 0.322 | 0.177 | 0.277 | 0.225 | 0.282 | 0.320 | 0.354 | 0.628 | 0.633 |
| SR and NPP | 0.331 | 0.178 | 0.287 | 0.195 | 0.209 | 0.176 | 0.312 | 0.303 | 0.643 | 0.497 |
| SE and WY | 0.205 | 0.245 | 0.208 | 0.220 | 0.237 | 0.252 | 0.231 | 0.214 | 0.507 | 0.522 |
| SE and NPP | −0.142 | −0.093 | −0.101 | −0.005 | 0.177 | 0.108 | 0.143 | 0.038 | 0.524 | 0.465 |
| NPP and WY | 0.579 | 0.631 | 0.549 | 0.582 | 0.755 | 0.747 | 0.917 | 0.900 | 0.997 | 0.96 |

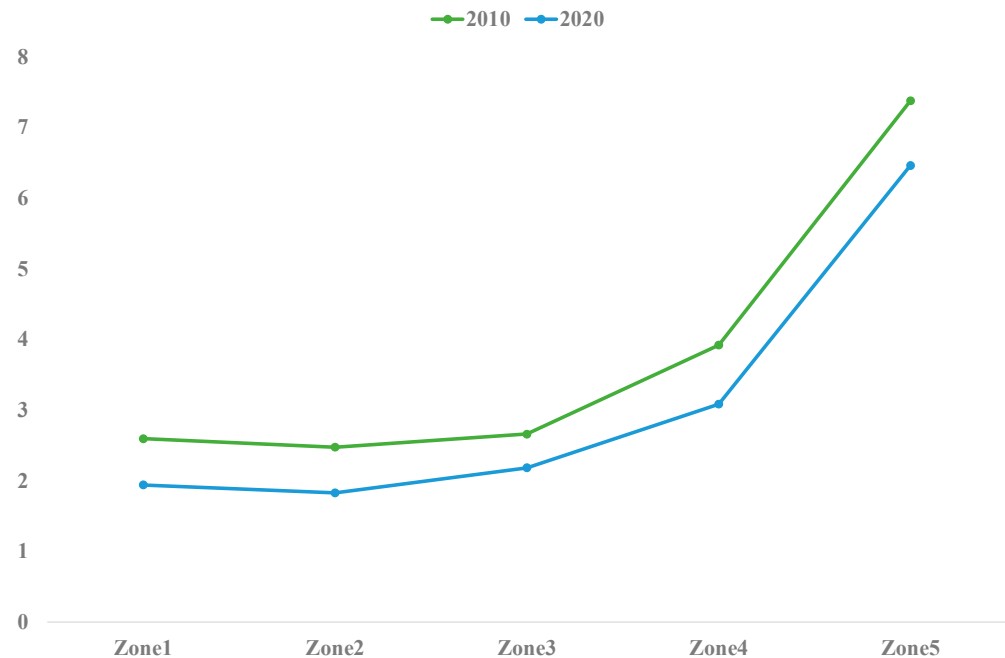

**Figure 9.** Evolutionary trend chart of the ES correlation coefficient sum of tea garden yield zoning in Fuzhou.

## 4. Discussion

### 4.1. Ecological Tea Gardens Construction Is Helpful to Achieve ESs Synergy

As the main crop of agricultural economic development in southern mountainous areas, tea has attracted wide attention [70]. There is a good overall facilitating relationship between ecosystems within tea gardens [71]. The $C_{ts}$ between all ESs in Fuzhou (Figure 5) have a synergistic relationship in the tea gardens and a trade-off relationship in the ETG areas, indicating that the ecological environment within the tea gardens is better than that in other areas. However, the synergistic relationship in the tea gardens had a decreasing trend from 2010 to 2020, which suggests an increased complexity of the ecological environment in the tea gardens.

Based on the results of the tea garden clustering division in Section 3.3, a corresponding transfer matrix for the transformation of different areas in 2010–2020 was created (Figure 10). In 2010, 1485.18 ha in Area I (Traditional Tea Garden Area) was transformed into Areas IV and V, accounting for approximately 96.72% of the Area I (Figure 11). These results show a good indication of ecological tea garden construction in Fuzhou in the past ten years. In addition, 1917.09 ha of Area IV (Mature Ecological Tea Garden Area) was converted to Area V (Immature Ecological Tea Garden Area) in 2010, accounting for approximately 97.25% of the Area IV (Figure 11).

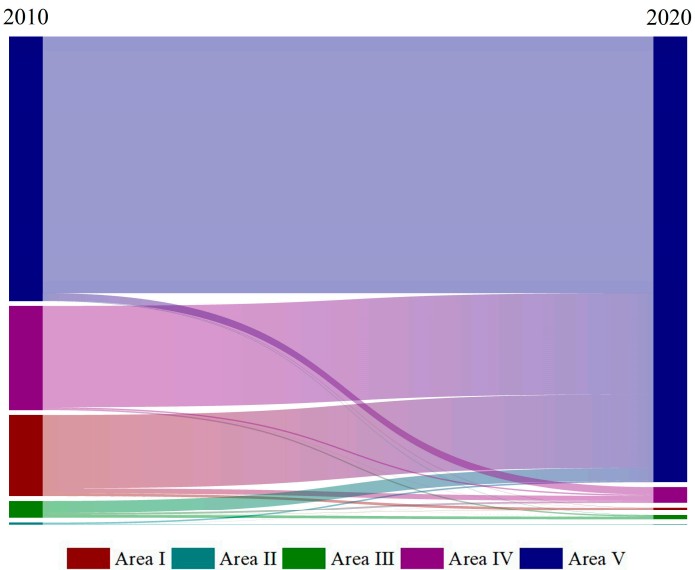

**Figure 10.** Sankey diagram of tea gardens cluster transfer from 2010 to 2020. This colorful bars represent the flow of different cluster areas of tea gardens in Fuzhou from 2010 to 2020.

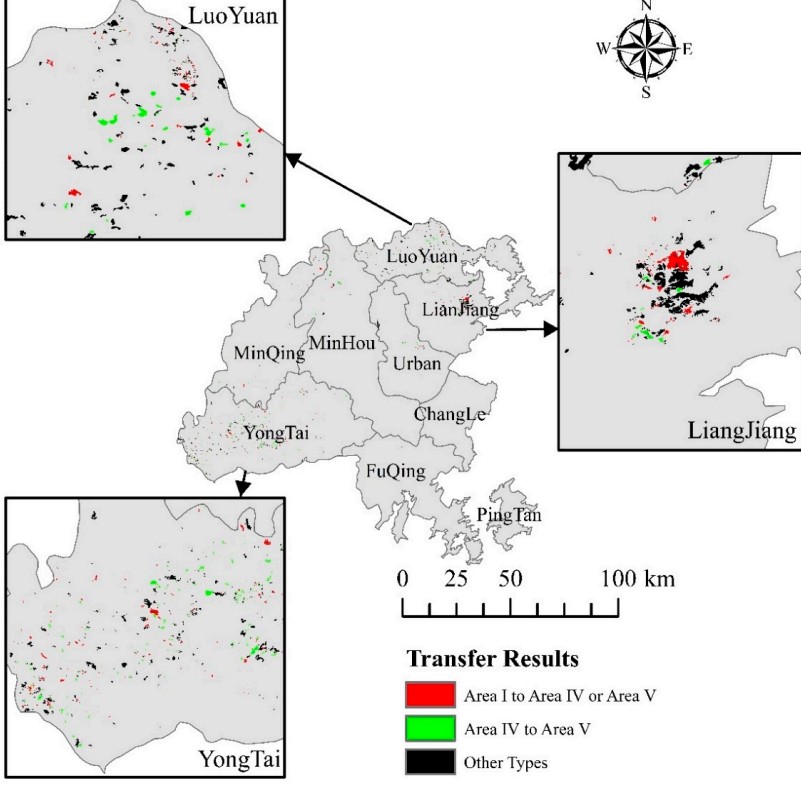

**Figure 11.** Example of tea garden cluster transfer results.

*4.2. Synergy among ESs Promotes Tea Production*

There are many evidences suggesting trade-off relationships between crop yield and ESs, especially regulation services such as increasing crop yield, leading to a decrease in NPP and SR capacity [72–74]. As a cash crop that depends on the ecological environment, tea shows a strong synergistic relationship between its output and ESs (Figure 9), which is consistent with the results of most other studies on agricultural ESs evaluations. Shui [75] noted that the positive service value of three service functions, supply, regulation and support, in the agricultural ecosystem dominated by specialized tea cultivation was greater than the negative service value, and the system mainly provided positive services. Compared with other crop ecosystems, specialized tea planting has a less negative impact on agricultural ecosystems [76]. Moreover, the synergistic relationships between tea production, WY and NPP can also be reflected through ecological tea garden management (Figure 5).

By further examining the trade-off/synergy relationship among ESs divided according to tea production, the $C_{ts}$ in each zone was positive (Figure 9), showing an overall synergistic relationship, and with the increase in tea yield, the synergistic relationship accelerated. It can be considered that the development of synergy among ESs in tea gardens can promote tea production. According to the trade-off/synergy among ESs, in the future tea garden governance process, we can focus on ESs with synergy, strengthen the ecological management of tea gardens and promote the increase in tea production and the governance of ecological tea gardens.

*4.3. The Decline of $C_{ts}$ Influenced by Climatic Factors*

According to the results of this study in Section 3.2, the $C_{ts}$ of tea gardens and ETG areas (Figure 6) show a significant difference. The $C_{ts}$ were positive in tea gardens and negative in ETG areas. However, $C_{ts}$ in both areas show a decreasing trend from 2010 to 2020. Figure 5a shows a strong synergistic relationship between WY and other ESs in the correlation analysis of tea gardens in 2010. CR also has a strong synergistic relationship with other ESs. However, this synergistic relationship changed significantly in 2020. Most of the synergistic correlations between ESs became smaller, or even changed to trade-off relationships (Figure 5b). It is the change in the correlation between WY and CR with other ESs that causes the decrease in $C_{ts}$. Our analysis of the existence of this phenomenon may be due to changes in climatic conditions in Fuzhou City in 2010 and 2020. The reason for these results is the extreme precipitation event in Fuzhou in 2010 and the strong influence of climatic factors on WY and CR [77]. Both WY and CR are higher in 2010 compared to the values in 2020. In future studies, we will search for more in depth reasons for the decline of $C_{ts}$ and its connection with climate factors.

*4.4. Recommendations for the Overall Management of the Ecological Tea Garden*

To promote the ecological environmental protection and sustainable development of tea gardens, based on the theory of ecosystem zoning management [78], our suggestions are as follows: (1) This study found that synergistic relationships among ESs contributed to tea yield. In the process of ecological tea gardens, the focus should be on ESs with synergistic effects. Increase tea production by enhancing the synergistic effects among ESs. (2) Our results supported the transition from traditional tea gardens to ecological tea gardens. The actual situation of the tea plantation land and the surrounding ecological environment should be fully considered in the construction process of ecological tea gardens to improve the enthusiasm for ecological management in the tea gardens and reduce environmental pollution on the basis of ensuring the maximum yield [79,80]. (3) Our results highlighted the importance of maintaining the stability of existing ecological tea gardens. The synergy between tea production and ESs should be balanced. While pursuing the construction of ecological tea gardens, attention should be given to the environmental management of the areas around the tea gardens to improve the stability of ecological tea gardens [81,82].

However, there are aspects of this study that need to be improved. Because it is difficult to obtain more accurate tea production data within the study area, only tea production data for 2010 and 2020 for each county in Fuzhou were selected for analysis in this study. In subsequent studies, attempts can be made to determine if more accurate data can be obtained for analysis.

## 5. Conclusions

Correlation and cluster analyses of trade-offs and synergies between ecosystem services within the tea gardens of Fuzhou revealed spatial structures and decadal changes that significantly affected tea yields. We defined $C_{ts}$ as the sum of the coefficients of trade-offs and synergies between ESs, to discover the trade-off/synergy in tea gardens and areas except tea gardens and explore the potential effects of synergy on tea yield. From 2010 to 2020, the bulk of Traditional Tea Garden Areas (Area I) were ecologically transformed into Mature Ecological Tea Gardens (Area IV) and Immature Ecological Tea Gardens (Area V). Meanwhile, the synergistic relationship of tea garden ESs promote tea production. These findings provide an idea for demonstrating the results of ecological tea gardens, as well as a method for studying the trade-off/synergy relationship of ESs in the region to assist in future management using approaches developed in this study.

**Author Contributions:** All the authors contributed significantly to this study. C.W., writing—original draft, methodology, conceptualization and software; M.Z., funding acquisition, methodology, resources, conceptualization; M.Z., Y.X., Y.Z. and X.Z., writing—review and editing. All authors have read and agreed to the published version of the manuscript.

**Funding:** This research was financially supported by the National Natural Science Foundation of China (Project Grant No. 42001217) and the State Key Laboratory of Earth Surface Processes and Resource Ecology, Beijing Normal University (No. 2021-KF-11).

**Data Availability Statement:** The data that support the findings of this study are available from the corresponding author, upon reasonable request.

**Conflicts of Interest:** The authors declare no conflict of interest.

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
