# Peer review of "Ecosystem Service Synergies Promote Ecological Tea Gardens: A Case Study in Fuzhou, China"

_remotesensing, doi:10.3390/rs15020540_

Round 1

Reviewer 1 Report

  1. The concept of ecosystem service in the MS should be reconsidered, for example, NPP itself is not an ecosystem service. NPP is a proxy for ES flow or when you say “Researchers have been attempting to determine the impact of ESs on crop production”
  2. Consider rewriting the methodology
  3. The method of estimating LAI is not clear
  4. The data used for Water yield modelling is not clear
  5. The data used for Soil retention modelling is not clear. And Why you used the P values from Ethiopia and India for a case study in China (references No. 37 and No.38), how about other parameters?
  6. The data used to get NPP is not clear
  7. Errors in referencing
  8. Figure 2 is too small
  9. The explanation of trade off is not clear

Reviewer 2 Report

Thank you for the opportunity to review your manuscript, which I know must have occupied a significant amount of time and effort. Unfortunately, I cannot recommend publication of the manuscript in its present state due to a number of significant difficulties with methodology, presentation, and interpretation of results. Overall, the manuscript must be made more reader-friendly so that it flows seamlessly in explaining the study, the data, methods, analyses, results and discussion/conclusions drawn from the results. Please refer to the attached document for detailed comments/suggestions. Please take the time to make the necessary modifications and revisions.

Reviewer 3 Report

Aiming to promote sustainable development of agroecosystems, the manuscript, taking tea gardens in Fuzhou as example, explored the trade-off / synergy among ecosystem services from supply services, regulation services, and ecosystem disservices. The study will provide a reference for government to construct ecological tea gardens. The method used in this manuscript is appropriate, the data are abundant, and the results and discussions are reasonable. There are several comments on the manuscript:

1          What non-tea garden areas refer to? You give several classifications of land use types in “Methods” section. In soil retention subsection, it includes “Plow, Forest, Lawn, Wetland, Water, Artificial Surface, Bare Land” in Table 1. In Carbon emissions subsection, it includes artificial surfaces, cultivated land, tea gardens, forests, grasslands, water bodies, sea areas, and unused land. This somewhat confuses the readers. Please explain them clearly.

2          I strongly suggested add a map or modified Figure 1 to add a classification of non-tea gardens.

3          Please give more discussion about why Cts in tea garden areas shows a declining characteristic from 2010 to 2020.

4          Several cross-references were showed that “Error! Reference source not found. “. Please check them.

Reviewer 4 Report

This paper mainly discusses the trade-offs/synergies between ecosystem services (ESs) of agroecosystems. The dissertation research only uses part of the products obtained by remote sensing technology and does not discuss new methods or new discoveries of remote sensing technology itself. The topic selection of the article does not meet the scope of the journal's topic selection, the topic selection is not novel, the paper is not innovative enough, no new methods or new insights are proposed, and the conclusions obtained by the article are not general and universal.

Round 2

Reviewer 2 Report

Minor comments/suggestions and a suggested Conclusion are in the attached report. Thank you to the authors for their diligent efforts.
